## Effects of antihypertensives, lipid-modifying drugs, glycaemic control drugs and sodium bicarbonate on the progression of stages 3 and 4 chronic kidney disease in adults: a systematic review and meta-analysis

Kathryn S Taylor,[1] Julie Mclellan,[1] Jan Y Verbakel,[1,2] Jeffrey K Aronson,[1] Daniel S Lasserson,[1,3] Nicola Pidduck,[1] Nia Roberts,[4] Susannah Fleming,[1] Christopher A O'Callaghan,[5] Clare R Bankhead,[1] Amitava Banerjee,[6] FD Richard Hobbs,[1] Rafael Perera[1]

**Correspondence to**
Julie Mclellan;
julie.mclellan@phc.ox.ac.uk

## ABSTRACT

**Objective** To evaluate the effects of drug interventions that may modify the progression of chronic kidney disease (CKD) in adults with CKD stages 3 and 4.

**Design** Systematic review and meta-analysis.

**Methods** Searching MEDLINE, EMBASE, Database of Abstracts of Reviews of Effects, Cochrane Central Register of Controlled Trials, Cochrane Database of Systematic Reviews, International Clinical Trials Registry Platform, Health Technology Assessment, Science Citation Index, Social Sciences Citation Index, Conference Proceedings Citation Index and Clinical Trials Register, from March 1999 to July 2018, we identified randomised controlled trials (RCTs) of drugs for hypertension, lipid modification, glycaemic control and sodium bicarbonate, compared with placebo, no drug or a drug from another class, in ≥40 adults with CKD stages 3 and/or 4, with at least 2 years of follow-up and reporting renal function (primary outcome), proteinuria, adverse events, maintenance dialysis, transplantation, cardiovascular events, cardiovascular mortality or all-cause mortality. Two reviewers independently screened citations and extracted data. For continuous outcomes, we used the ratio of means (ROM) at the end of the trial in random-effects meta-analyses. We assessed methodological quality with the Cochrane Risk of Bias Tool and confidence in the evidence using the Grading of Recommendations Assessment, Development and Evaluation (GRADE) framework.

**Results** We included 35 RCTs and over 51 000 patients. Data were limited, and heterogeneity varied. Final renal function (estimated glomerular filtration rate) was 6% higher in those taking glycaemic control drugs (ROM 1.06, 95% CI 1.02 to 1.10, I²=0%, low GRADE confidence) and 4% higher in those taking lipid-modifying drugs (ROM 1.04, 95% CI 1.00 to 1.08, I²=88%, very low GRADE confidence). For RCTs of antihypertensive drugs, there were no significant differences in renal function. Treatment with lipid-modifying drugs led to a 36% reduction in cardiovascular disease and 26% reduction in all-cause mortality.

### Strengths and limitations of this study

► By focusing on patients with chronic kidney disease (CKD) stages 3 and 4, we provide a primary care perspective on the management of CKD.

► We completed an extensive and comprehensive search including common drug mis-spellings, with no restriction on language or publication type.

► Generalisability of our findings is limited by the paucity of trials, high statistical heterogeneity, high clinical heterogeneity, in terms of comorbidities and by the effect of drugs on CKD being secondary in all the studies.

► Lack of data restricted our ability to explore the baseline characteristics of studies, the effects of drugs, and the link between treatment effect and drug dose, and assess the potential for publication bias.

**Conclusions** Glycaemic control and lipid-modifying drugs may slow the progression of CKD, but we found no pooled evidence of benefit nor harm from antihypertensive drugs. However, given the data limitations, further research is needed to confirm these findings.

**PROSPERO registration number** CRD42015017501.

## INTRODUCTION

Chronic kidney disease (CKD) is a long-term loss of renal function and is a global health problem, with an estimated prevalence of 11%–13%.[1] CKD is an independent risk factor for cardiovascular disease,[2] and people with mild-to-moderate CKD are at increased risk of mortality and progression to renal failure.[3]

Historically, CKD was categorised by its cause, which was predominantly diabetes,

hypertension, glomerulonephritis or polycystic kidney disease. Treatment could be recommended, but often opportunities were missed for early detection of the disease or prevention of clinical complications and further deterioration. The National Kidney Foundation addressed this challenge by developing a five-stage classification system based on the glomerular filtration rate (GFR) (online supplementary material),[4] which was subsequently modified to take into account degrees of proteinuria (online supplementary material).[5] A UK audit in 2012 estimated that 90% of cases of CKD are at stage 3, and of these 84% are at stage 3a (estimated GFR (eGFR) 45–59 mL/min/1.73$^2$).[6]

Patients with CKD who are managed in primary care include those in stages 1 and 2, with an emphasis on lifestyle and diet changes. Patients in stages 3 and 4 may be managed either in primary care or shared care with specialist nephrology services, with a greater use of medication, primarily for associated long-term conditions, such as diabetes and hypertension. Both published data for Oxfordshire[7] and unpublished national data from our group show that some patients with CKD stage 4 have their renal function monitored in primary care.

Four different classes of drugs may modify CKD progression: antihypertensives,[8] lipid-modifying drugs,[9] glycaemic control medications in patients with diabetes[10] and sodium bicarbonate.[11–13] Sodium bicarbonate addresses metabolic acidosis, a prevalent complication in moderate and late-stage CKD, and may offer a simple and low-cost treatment to slow CKD progression. The effects of drugs on CKD are secondary, as these drugs are primarily used to treat hypertension, hypercholesterolaemia or diabetes.

Systematic reviews have been compiled to summarise the effects of drug interventions on the rate of CKD progression.[9 10 14–16] However, while two have reported results based on eGFR, none has reported results with reference to CKD staging. Furthermore, no systematic review has compared the effects of different drug classes.

Our objective was to carry out a systematic review and meta-analysis of randomised controlled trials (RCTs) to examine and compare the effects on the progression of CKD of the four classes of drugs, focusing on patients who can be managed in primary care or shared care with specialist nephrology services.

## METHODS

Our systematic review protocol was registered with PROSPERO (International Prospective Register of Systematic Reviews) and reported in line with the recommendations from the Preferred Reporting Items for Systematic Reviews and Meta-Analyses statement.[17]

### Search strategy

The search strategy was designed by an information specialist (NR) with advice from clinicians including nephrologists. Medical Subject Headings (MeSH) terms included terms for population, study design and intervention drugs (randomised control trials, chronic kidney, renal, drug classes and specific drug names, including common mis-spellings). Our included drug list is shown in the online supplementary material.

Searches were conducted in Ovid MEDLINE (online supplementary material), Cochrane Central Register of Controlled Trials, Cochrane Database of Systematic Reviews, Database of Abstracts of Reviews of Effects, EMBASE, Health Technology Assessment, Science Citation Index, Social Sciences Citation Index, Conference Proceedings Citation Index, Clinical Trials Register, and the WHO International Clinical Trials Registry Platform to 6 July 2018 to identify relevant studies, regardless of their publication type. No language restriction was applied. All publications from March 1999 onwards were considered, to align with the publication of the first equation to calculate eGFR, which had led to the introduction of CKD staging.[18] Reference lists were also examined.

### Study selection

Two reviewers (KST, JM, NP and SF; in pairs) reviewed the title and abstract of each reference and identified potentially relevant references. Considering the full texts of these studies, two reviewers independently selected studies to be included in the review using predetermined inclusion criteria. Disagreements about study inclusion were resolved by a third reviewer.

The following were our study inclusion criteria:
1. RCTs of drugs that may modify the progression of CKD: antihypertensives, lipid-modifying drugs, glycaemic control drugs and sodium bicarbonate.
2. Control arms were either given a placebo drug, no drug intervention or a comparator drug from one of the three other classes.
3. Adult participants.
4. Patients with stage 3 and/or stage 4 CKD. Studies additionally including patients with stage 1 and/or stage 2 CKD were also included to increase the number of potential studies, but we would expect the treatment effect to be overestimated in these studies, as patients with stage 1 and 2 CKD would respond better to treatment.
5. Populations with at least 40 participants.
6. Follow-up of at least 2 years.
7. Publications reporting a quantitative summary of effect as a change from baseline, or an endpoint measure of at least one of seven prespecified outcomes.
   i. Renal function, which could be measured by GFR, eGFR, creatinine clearance (CrCl) or estimated CrCl. This was our primary outcome of interest.
   ii. Proteinuria, which could be measured by protein excretion rate (PER), protein creatinine ratio (PCR), albumin excretion rate (AER) or albumin creatinine ratio (ACR).
   iii. Adverse events.
   iv. Commencement of maintenance dialysis or kidney transplantation.
   v. Cardiovascular events.

vi. Cardiovascular mortality.

vii. All-cause mortality.

We excluded trials that reported only data about populations that included people with CKD stage 5 to avoid underestimating the treatment effect, as we would expect stage 5 patients to have more comorbidities, which might lead to worse outcomes.

## Data extraction and quality assessment

Data were extracted independently by two reviewers (KST, JM, NP and JYV; in pairs) using a data extraction form, which was piloted on a sample of five studies.

Extracted data included study details (setting, intervention dose and frequency, follow-up period), information on study participants (age, gender, ethnicity, smoking status, blood pressure, CKD stage, existing comorbidities) and outcome data. For one study[19 20] we extracted endpoint eGFR data from an earlier point than the longest follow-up as the patient numbers at the longest follow-up were so small (<7% of the population). We contacted study authors to seek clarification about data and to request data that could be included in our study.

The methodological quality of studies was assessed independently by two reviewers (KST, JM, NP and JYV; in pairs) using the Cochrane Risk of Bias Tool.[21] This tool considers the method of random sequence generation and allocation concealment (selection bias), blinding of participants, personnel and outcomes (performance and detection biases), the levels and balance of missing outcome data (attrition bias), and completeness of reporting (selective reporting bias).

Disagreements about extracted data or methodological assessments were resolved by a third reviewer.

## Data synthesis and analysis

We expected high statistical heterogeneity,[22] but pooled data to give an indication of the average effect. We also pooled data when there were only two or three studies, although this could be potentially misleading, in order to give an indication of potential trends had there been more study data. We carried out random-effects analyses based on the DerSimonian and Laird method,[23] assessed statistical heterogeneity using the $I^2$ statistic and calculated approximate 95% prediction intervals to estimate the likely effects in a clinical setting[24] using the methods of Higgins *et al*.[25]

For the continuous outcomes (renal function and proteinuria), we used a ratio of means (ROM)[26 27] effect measure, which allowed us to combine data from trials that reported different measures of renal function or proteinuria in a single analysis. Using the ROM effect measure overcomes limitations of the well-established standardised mean difference (SMD),[21] by providing units that are more easily interpreted than SDs, and avoiding the problems associated with presenting results with SDs, which can produce misleading results by deflating treatment effects for studies of heterogeneous populations and inflating treatment effects in homogeneous populations.[28] We calculated the ROM as the ratio of the final eGFR in the intervention group to the final eGFR in the comparator group, so ROM >1 favoured the intervention group for eGFR, where higher values are better than lower values. If a study of renal function reported reductions from baseline or rates of decline rather than endpoint data, we estimated the ROM of endpoints by the ratio of the reduction from baseline or rate of decline in the comparator group to that in the intervention group (the reciprocal) to ensure the direction of effect was consistent across all studies, as higher values of declining renal function are worse. ROM <1 favoured the intervention group for the proteinuria outcomes (ACR, AER, PER, PCR), where higher values are worse.

For a few studies it was necessary to make further estimates by approximating a mean using the median; estimating SD from the IQR[21]; or estimating SE of ROM by imputation using the mean value of the known SEs of the other studies.

Meta-analysis was carried out on the log-transformed ROM and its SE. For the primary analysis, eGFR measured by the Modification of Diet in Renal Disease (MDRD) equation was selected over Chronic Kidney Disease Epidemiology Collaboration (CKD-EPI) data, as MDRD is the main equation advocated in clinical guidelines and CKD-EPI was not introduced until 2009[5 29]; eGFR was also selected over CrCl. When analysing the effects of treatments on proteinuria, we pooled data for PER, PCR, ACR and AER using a ROM effect measure. For the dichotomous outcomes, we specified the relative risk (RR) as the effect measure.

We performed separate sensitivity analyses to ensure that our estimates of the effect of treatment in studies of patients with CKD stages 3 and 4 only (our main focus) were not sensitive to our choice of model, assumptions, quality of the studies, estimations made or our choices of data.

The following were the prespecified sensitivity analyses:

► Fixed-effect analyses based on the inverse-variance method were carried out for the continuous outcomes and based on the Mantel-Haenszel method for the dichotomous outcomes.[30 31]

► Excluding studies rated as low quality, which we defined as having a high risk of attrition bias or allocation concealment bias; studies rated as 'unclear' were not excluded.

► Excluding studies without intention-to-treat analysis. This was prespecified on the assumption that we would be dealing with primary studies of the progression of CKD, but as the studies were secondary or post-hoc this analysis was no longer relevant.

The following were further, standard set of sensitivity analyses:

► Excluding studies with estimates.

► Using more conservative estimates, considering those of the SE of ROM by imputing with the 75% percentile value of the known SEs rather than the mean value, reflecting greater uncertainty.

► Using alternative data, considering CKD-EPI data instead of MDRD data, for studies that measured eGFR using both equations.

► Excluding studies that measured PCR and PER in the analysis of proteinuria, as urine albumin is a more accurate measure of glomerular damage.

► Excluding each individual study, one by one, to see if the level of heterogeneity depended on a particular study.

The following were post-hoc sensitivity analyses:

► Using the Hartung-Knapp-Sidik-Jonkman variance correction to calculate 95% CIs reflecting the uncertainty in heterogeneity estimates.[32–34]

► Excluding studies with non-placebo controls.

Our prespecified subgroup analyses were by age (<65 or ≥65 years), ethnicity, smoking status, intensity of intervention, high blood pressure (≥140/90 mm Hg), low blood pressure (≤90/60 mm Hg), method of eGFR measurement, degree of proteinuria in the renal function analysis, drug type within each drug class and by CKD stages 3 and 4 separately. Heterogeneity was assessed using the $I^2$ statistic. Publication bias was explored using funnel plot analysis with contour plots.[21 35]

We used the Grading of Recommendations Assessment, Development and Evaluation (GRADE) framework[21] to report the overall quality of evidence for the primary outcome (final eGFR). The certainty in the evidence for each outcome was graded as high, moderate, low or very low.

When describing the included trials, we refer to 'studies' to encompass single trials and publications that report more than one trial. When reporting analyses of the effects of the drugs, we refer to populations as some studies reported data for CKD subgroups or dose-dependent subgroups. All results of meta-analysis are displayed in forest plots and/or tabulated. Results are divided into those for studies of patients with CKD stages 3 and 4 only and studies of patients with CKD stages 3 and 4 mixed with patients with stages 1 and 2.

All analyses were carried out using STATA v.14.2,[36] except the risk of bias figures which were created in RevMan v.5.3.[37]

### Patient involvement
Members of a patient and public involvement group were part of the Stakeholder Group and Steering Committee of the National Institute for Health Research (NIHR) Programme Grant that inspired this study. No patients or patient representatives were involved in setting the outcome measures, nor were they directly involved in developing plans for the design or implementation of the study. Three members of this group commented on our manuscript and we thank them for their help in the Acknowledgement section. A 1-day dissemination event is planned to report the results of all the studies funded by the NIHR Programme Grant, including this study. Members of the patient and public involvement group will be invited to this event.

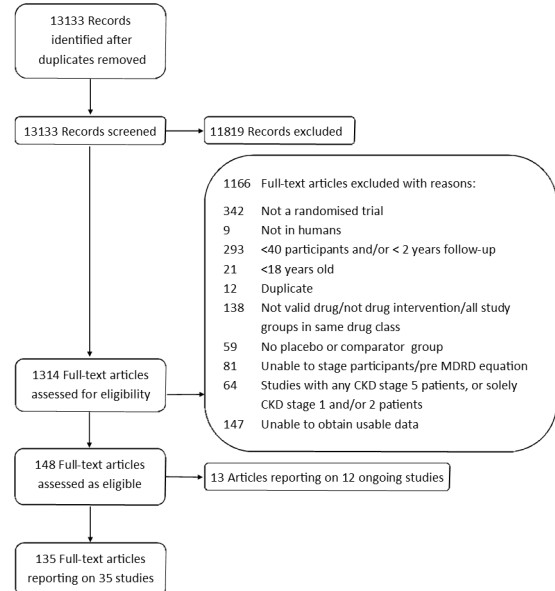

**Figure 1** PRISMA diagram. PRISMA, Preferred Reporting Items for Systematic Reviews and Meta-Analyses. CKD, chronic kidney disease; MDRD, Modification of Diet in Renal Disease.

## RESULTS
### Search results
Through database registry and hand searches, we identified 13 133 records (figure 1), of which 35 studies (51 155 patients) satisfied our eligibility criteria.

### Study characteristics
Most of the 35 studies reported data about a single trial, while 3 studies reported data about a pooled analysis of more than one trial.[19 20 38–43] Of the 35 studies, 1 was from China,[44] 1 from Tasmania,[45] 3 from Japan,[46–51] 6 from USA/Canada,[52–65] 9 from Europe,[13 66–78] either in single or multiple countries, and the other 15 studies were multinational (online supplementary material). Nineteen studies[13 19 20 38–40 44 48–57 61–65 68–70 74 75 78–90] provided data for populations with CKD stages 3 and/or 4 only. These studies had between 108 and 3094 patients, and the mean or median follow-up was between 24 and 66 months.

One study[67] had a 2×2 factorial design involving an antihypertensive drug and a lipid-modifying drug, and this study is included in our analysis of both antihypertensive drugs and lipid-modifying drugs. Another study had a 2×2[71 72] factorial design involving two antihypertensive drugs and a lipid-modifying drug, and this study is only included in our analysis of lipid-modifying drugs because all patients received antihypertensives. The other 33 studies were 12 studies of antihypertensives,[16 38–40 44 46 47 52–57 66 67 79–83 91–98] 13 of lipid-modifying drugs,[45 48–51 58–63 68–70 73–78 84–86 99–104] 6 of glycaemic control agents[19 20 41–43 87–90 105–107] and 2 of sodium bicarbonate[13 64 65] (online supplementary material). Four studies[45 58–60 73 102–104] had populations with CKD and no comorbidities highlighted, 27 studies had a single comorbidity, and 4 studies had more than one

The flow diagram contains the following:

13133 Records identified after duplicates removed

13133 Records screened → 11819 Records excluded

1166 Full-text articles excluded with reasons:
- 342 Not a randomised trial
- 9 Not in humans
- 293 <40 participants and/or < 2 years follow-up
- 21 <18 years old
- 12 Duplicate
- 138 Not valid drug/not drug intervention/all study groups in same drug class
- 59 No placebo or comparator group
- 81 Unable to stage participants/pre MDRD equation
- 64 Studies with any CKD stage 5 patients, or solely CKD stage 1 and/or 2 patients
- 147 Unable to obtain usable data

1314 Full-text articles assessed for eligibility

148 Full-text articles assessed as eligible → 13 Articles reporting on 12 ongoing studies

135 Full-text articles reporting on 35 studies

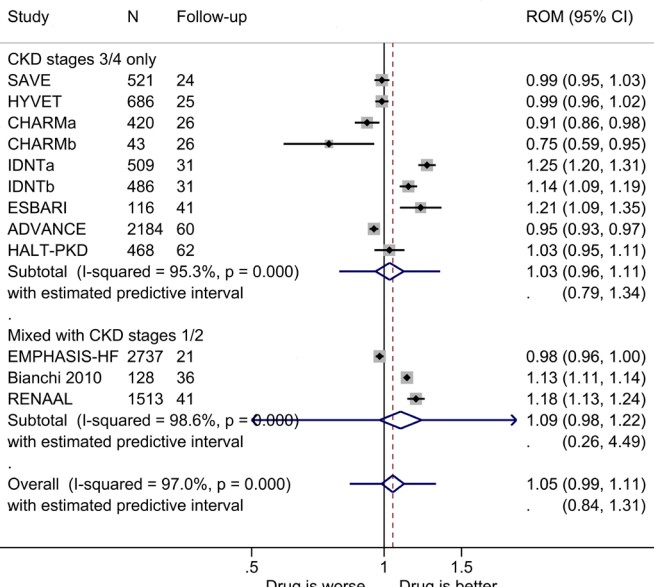

**Figure 2** ROM of estimated glomerular filtration rate at the end of the trials for antihypertensives versus comparator (boxes) and pooled estimates across studies (diamonds) calculated by the random-effects DerSimonian and Laird method, split by CKD stage. CHARMa and CHARMb, CKD stages 3 and 4, respectively, with estimated glomerular filtration rate measured by the MDRD equation. IDNTa, intervention is irbesartan; IDNTb, intervention is amlodipine. CKD, chronic kidney disease; MDRD, Modification of Diet in Renal Disease; ROM, ratio of means.

comorbidity. Comorbidities were cardiovascular in 19 studies[19 20 38–43 46–51 53 54 57 61–63 68–72 74–78 81–83 89–92 98–101]; type II diabetes in 9 studies[19 20 41–43 53–56 79 80 87–90 93–97 105–107] (including all 6 studies of glycaemic control drugs); and a renal disorder or damage (idiopathic chronic glomerulonephritis, advanced chronic renal insufficiency without diabetes, advanced autosomal dominant polycystic kidney disease, renal impairment and metabolic acidosis) in 7 studies[13 44 52 64–67 84–86] (online supplementary material). In 15 studies,[13 45–47 61–67 71–73 76 77 91–99 107] the population analysed for our primary outcome was the whole trial population, and in the other 20 the population analysed was a subgroup of the trial population, reported as either a subgroup or post-hoc analysis (online supplementary material).

Baseline data were provided for the same population as that included in our analyses for 19 of the 35 studies[13 45 49–51 58–70 73 76–80 84–88 91–98 107]; for the other 16 studies, baseline data were only provided for either a different subset of the trial population or the whole trial population (online supplementary material).

### Risk of bias

We considered the methodological quality of all 35 eligible studies (online supplementary material), based on the design of the trial(s) from which data were obtained. Eight studies[19 20 44 71–73 84–88 98 105 106] were ranked with high quality as they had 'Low' risk of bias across all domains, and we rated one study[76 77] as 'High' risk of bias

for three domains and 'Unclear' across two domains. Most studies described their method of randomisation, but fewer studies were clear about how they concealed the group allocation and whether participants, personnel and outcome assessors were blinded. The risk of attrition bias was high in nearly a quarter of the studies.

We had sufficient data to consider the possibility of publication bias based on the primary outcome of renal function in those taking antihypertensive and lipid-modifying drugs. The contour-enhanced funnel plots suggested possible publication bias, as few studies were in the area of non-significance (online supplementary material).

### Renal function

Most of the available data were on eGFR.

The pooled ROM of eGFR for antihypertensive drugs versus comparators at the end of the studies is shown in figure 2. There was no significant effect in studies of populations with CKD stages 3 and 4, and heterogeneity between studies was substantial at 94% (yielding a prediction interval of 0.79–1.34), and this was not attributable to a single study. Adding data from the studies of patients that included those with CKD stages 1 and/or 2 increased the benefit to the intervention group but not significantly. All except one study (Bianchi *et al*[66]) had placebo controls. All the sensitivity analyses produced similar results of no significant treatment effect and heterogeneity remained substantial (online supplementary material).

The only subgroup analysis that was possible, of those prespecified, was by drug type within the antihypertensive drug class. Among studies of CKD stages 3 and 4 only, there were no significant treatment effects within the different drug types (online supplementary material), except in the case of calcium channel blockers, where the endpoint eGFR was 14% higher in the intervention group (ROM 1.14, 95% CI 1.01 to 1.27), but this was based on a single study.[53 54]

In studies of patients with CKD stages 3 and 4, there was a marginally higher benefit to those taking the lipid-modifying drugs (figure 3), as the endpoint eGFR was 4% higher (ROM 1.04, 1.00 to 1.08, statistically significant, $I^2$=88; STATA's metaninf command produced output that indicated that the lower confidence bound was 1.0001 to 4 decimal places) with a prediction interval of 0.91–1.18. This result was not robust as the results of sensitivity analysis were variable (online supplementary material). The high degree of heterogeneity was attributed to the Study of Heart and Renal Protection[102–104] of the CKD stage 4 subgroup, as removing this study reduced $I^2$ from 88.3% to 45.3%. The intervention for this study was a higher dose of statin than in the other studies, and another lipid-modifying drug, ezetimibe, was also used. Adding data from the studies of patients including those with CKD stages 1 and/or 2 increased the benefit to the intervention group but not significantly, and there was substantial heterogeneity between studies ($I^2$=98%). Three trials had non-placebo controls (ASUCA,[48] MEGA[49–51] and GREACE[76 77]).

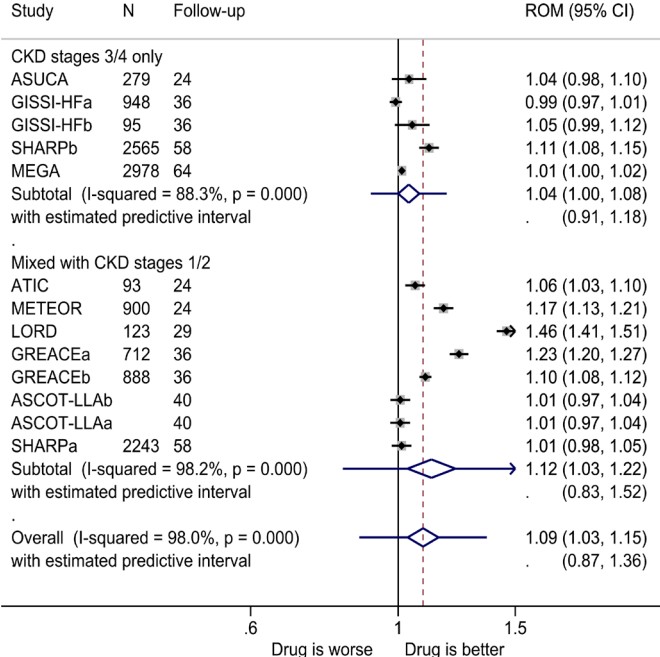

**Figure 3** ROM of estimated glomerular filtration rate at the end of the trials for lipid-modifying drugs versus comparator (boxes) and pooled estimates across studies (diamonds) calculated by the random-effects DerSimonian and Laird method, split by CKD stage. ASCOT-LLAa, atorvastatin and amlodipine versus placebo and amlodipine; ASCOT-LLAb, atorvastatin and atenolol versus placebo and atenolol; GISSI-HFa, CKD stage 3; GISSI-HFb, CKD stage 4; GREACEa, atorvastatin versus usual care for the group with metabolic syndrome; GREACEb, atorvastatin versus usual care for the group without metabolic syndrome; SHARPa, CKD stages 1–3; SHARPb, CKD stage 4. CKD, chronic kidney disease; ROM, ratio of means; SHARP, Study of Heart and Renal Protection.

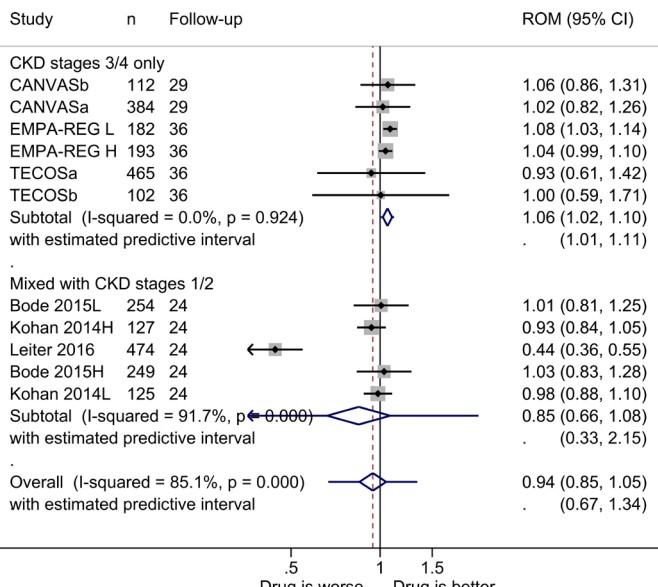

**Figure 4** ROM of estimated glomerular filtration rate at the end of the trials for glycaemic control drugs versus comparator (boxes) and pooled estimates across studies (diamonds) calculated by the random-effects DerSimonian and Laird method, split by CKD stage. Bode 2015L, intervention is low-dose canagliflozin (100 mg per day); Bode 2015H,intervention is high-dose canagliflozin (300 mg per day); EMPA-REG H, intervention is high-dose empagliflozin (25 mg daily); EMPA-REG L, intervention is low-dose empagliflozin (10 mg daily); Kohan 2014L, intervention is low-dose dapagliflozin (5 mg per day); Kohan 2014H, intervention is high-dose dapagliflozin (10 mg per day); CANVASa, CKD stage 3a; CANVASb, CKD stage 3b; TECOSa, CKD stage 3a; TECOSb, CKD stage 3b. CKD, chronic kidney disease; ROM, ratio of means.

There was a significant benefit to those taking glycaemic controlling drugs (figure 4) as the mean eGFR at the end of the study was 6% higher in the intervention group (ROM 1.06, 1.02 to 1.10). Heterogeneity between studies was low ($I^2$=0%), yielding a prediction interval of 1.01–1.11. Sensitivity analyses showed that this result was robust (online supplementary material). Adding the CKD stage 1–4 studies, there was no significant difference between the intervention and comparator groups, and the heterogeneity between studies was high ($I^2$=85%). In all studies the comparator groups received placebo drugs. We assessed the outlier[41–43] as having high attrition bias, as of the 1887 who were randomised only 487 completed the 104-day study period. After excluding this study, there was a significant benefit to the intervention group, which was similar to that for the studies of patients with CKD stages 3 and 4.

A study[64 65] of patients with CKD stage 3 reported a 22% higher endpoint eGFR from taking sodium bicarbonate (ROM 1.22, 1.10 to 1.36). Another study[13] of patients with CKD stage 4 reported that taking sodium bicarbonate produced no significant effect on renal function in terms of CrCl (ROM 0.94, 0.85 to 1.05). Pooling these

data, there was no significant treatment effect based on renal function (online supplementary material). In both studies the comparator groups received routine care.

Figure 5 summarises the effects of treatment on renal function by drug class for the four drug classes.

The GRADE confidence in these estimates was very low for our analysis of antihypertensive drugs, lipid-regulating drugs and sodium bicarbonate, and low for our analysis of glycaemic control drugs.

### Proteinuria

Data pooling for proteinuria was possible only for antihypertensive drugs in five studies. For studies of CKD stages 3 and 4 and overall, there was no significant treatment effect (online supplementary material), with ROM of 0.91 (0.78 to 1.05) and low heterogeneity at 41.3%. All sensitivity analyses produced the same result of no treatment effect with low heterogeneity (online supplementary material).

Two studies of lipid-modifying drugs presented proteinuria. The ATIC study[73] of patients with CKD stages 2–4 reported that treatment reduced urinary albumin excretion and provided medians and ranges of the AER (77, 3–2509 mg/24 hours for the intervention group, and 107,

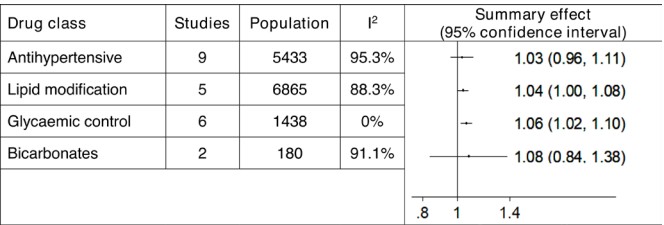

| Drug class | Studies | Population | I² | Summary effect (95% confidence interval) |
|---|---|---|---|---|
| Antihypertensive | 9 | 5433 | 95.3% | 1.03 (0.96, 1.11) |
| Lipid modification | 5 | 6865 | 88.3% | 1.04 (1.00, 1.08) |
| Glycaemic control | 6 | 1438 | 0% | 1.06 (1.02, 1.10) |
| Bicarbonates | 2 | 180 | 91.1% | 1.08 (0.84, 1.38) |

**Figure 5** Effect of treatment on estimated glomerular filtration rate by drug class: CKD stages 3 and 4 only. ROM >1 favoured the intervention group (higher values are better). 'Studies' refers to the study of different populations. Of the 35 included studies, 15 reported renal function of patients with CKD stage 3 and/or stage 4. Of these, 9 reported renal function of a single population and 6 reported results for two CKD subgroups or dose-dependent subgroups. CKD, chronic kidney disease; ROM, ratio of means.

5–3545 mg/24 hours for the comparator group). The PREVEND IT lipid study[67] of patients with CKD stages 1–3 reported that treatment slightly increased urinary albumin excretion by quoting medians and IQRs (21.8 and 11.6–41.9 mg/24 hours for the intervention group, and 20.3 and 12.5–40.5 mg/24 hours for the control group).

One glycaemic control study presented proteinuria as an outcome. The CANVAS programme[19 20] reported that treatment significantly reduced albuminuria compared with the placebo group. ACR was expressed on the geometric mean scale, as 13% lower in the intervention group compared with the controls (95% CI 1% to 24%)

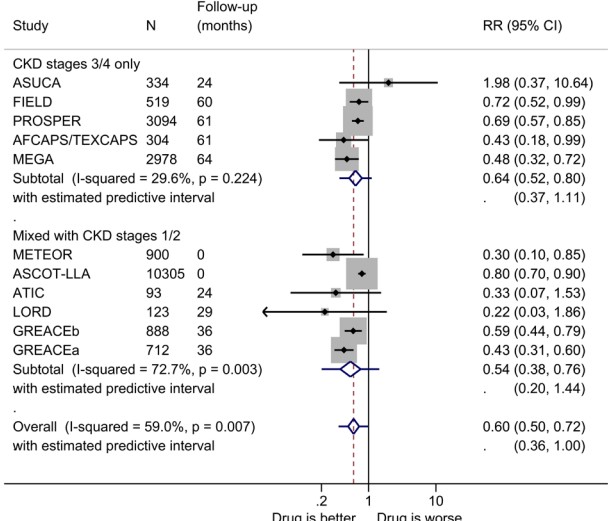

**Figure 6** Relative risk (RR) of cardiovascular disease during trials of lipid-modifying drugs versus comparator (boxes) and pooled estimates across trials (diamonds) calculated by the random-effects DerSimonian and Laird method, split by CKD stage. GREACEa, atorvastatin versus usual care for the group with metabolic syndrome; GREACEb, atorvastatin versus usual care for the group without metabolic syndrome. CKD, chronic kidney disease.

in patients with CKD stage 3a, and 26% lower (95% CI 20% to 31%) in patients with CKD stages 3b and 4.

One sodium bicarbonate study[64 65] reported that treatment lowered the ACR by quoting a median (IQR) of 257.1 (205.5–305.0 mg/g creatinine) for the intervention group and 300.3 (241.8–347.5 mg/g creatinine) for the control group.

### Adverse events
There were no significant differences in the risk of adverse events in the studies of antihypertensives, lipid-modifying drugs and glycaemic control agents (online supplementary material). Neither of the bicarbonate studies reported adverse events.

### Maintenance dialysis and kidney transplantation
Two studies of antihypertensive drugs reported the number of patients who deteriorated and required maintenance dialysis or kidney transplantation. The risk in the intervention group was 31% lower than in the control group in the HALT PKD study[52] of patients with stages 3 and 4 CKD (RR 0.69, 95% CI 0.50 to 0.96). In the RENAAL study,[93–97] which was based on patients with CKD stages 2–4, treatment with antihypertensives led to a 23% reduction in risk of deterioration to kidney failure (RR 0.77, 95% CI 0.64 to 0.93). There were no significant differences in this outcome in the studies of lipid-modifying drugs or glycaemic control drugs (online supplementary material).

### Cardiovascular events
There were no significant differences between the intervention and control groups in the risk of cardiovascular events in the studies of antihypertensives, whereas treatment with lipid-modifying drugs led to a reduction in risk of 36% (RR 0.64, 95% CI 0.52 to 0.80, I²=30%) (figure 6). All the sensitivity analyses yielded similar results and heterogeneity generally remained low (online supplementary material). The reduction in risk was slightly higher when data were added from the studies of patients that included those with CKD stages 1 and/or 2, but this reduction in risk was not significant (online supplementary material).

The TECOS glycaemic control study[89 90] reported the number of patients who experienced cardiovascular events in patients with CKD stage 3. They reported no significant differences between the intervention and control groups (online supplementary material).

### Mortality
In both our analyses of the studies of antihypertensive drugs and glycaemic control drugs, we found no significant differences, between the intervention and comparator groups, in the risk of cardiovascular mortality and all-cause mortality (online supplementary material).

Two studies reported the effect of lipid-modifying drugs on the risk of cardiovascular mortality in patients with CKD stages 3 and 4. In the FIELD study,[84–86] the risk was 47% lower in the intervention group (RR 0.53, 95% CI

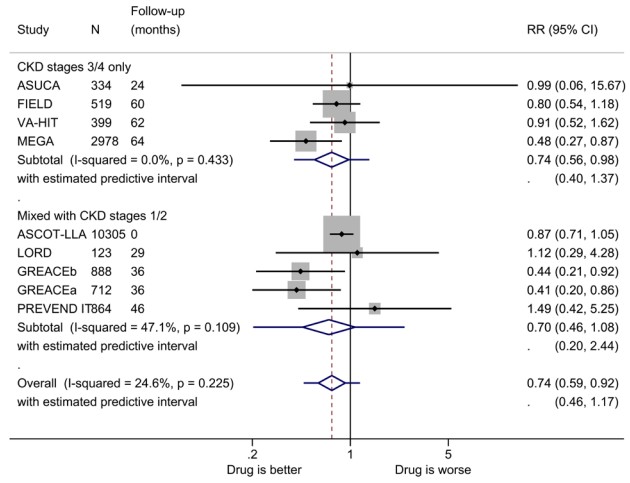

**Figure 7** Relative risk (RR) of all-cause mortality during trials of lipid-modifying drugs versus comparator (boxes) and pooled estimates across trials (diamonds) calculated by the random-effects DerSimonian and Laird method, split by CKD stage. GREACEa, atorvastatin versus usual care for the group with metabolic syndrome; GREACEb, atorvastatin versus usual care for the group without metabolic syndrome. CKD, chronic kidney disease.

0.30 to 0.92), and in the AFCAPS/TEXCAPS study[58–60] there was no significant difference.

In the studies of patients with CKD stages 3 and 4 only, treatment with lipid-modifying drugs reduced the risk of all-cause mortality by 26% (RR 0.74, 95% CI 0.56 to 0.98) (figure 7 and online supplementary material).

## DISCUSSION
### Main findings
We found limited data on the effects of antihypertensive, lipid-modifying and glycaemic control drugs and sodium bicarbonate on the progression of renal function in patients with CKD stages 3 and 4. There was heterogeneity of patient populations in the included studies. Pooling data provided no pooled evidence of benefit or impairment of renal function in these patients attributable to antihypertensive drugs. There was some suggestion of benefit to renal function from taking glycaemic control drugs and lipid-modifying drugs, and a clear reduction in cardiovascular disease and all-cause mortality by taking lipid-modifying drugs. There was insufficient evidence to draw any conclusions about sodium bicarbonate. There were only sufficient data to study the effects of antihypertensive drugs on proteinuria and we found no significant treatment effect.

### Comparison with other reviews
Our review is comparable with two reviews of drug interventions in patients with CKD, as they report changes in eGFR as outcome measures. Our findings on the protective effect of lipid-modifying drugs are consistent with the findings of Sandhu et al,[9] who concluded that statins

produce a small reduction in the progression of CKD in patients with cardiovascular disease. Our results are also consistent with those of Lewis et al,[16] who reported the preliminary results of a review that considered the effect on kidney function of lowering low-density lipoprotein cholesterol with statins. Their preliminary findings suggested that for patients taking statins there was a highly significant reduction in the annual rate of CKD progression compared with control groups (based on the annual rate of change in eGFR). Both of these studies considered general populations with CKD. Our finding that lipid-modifying drugs reduce the risk of cardiovascular events and all-cause mortality also concurs with other reviews of patients with CKD.[108–110]

### Strengths and limitations
Our study is a first attempt to summarise the literature on drug treatment for patients with CKD types 3 and 4. We have identified, evaluated and compared the effects of different drug interventions on reducing the rate of CKD progression in patients with CKD stages 3 and 4. By focusing on this patient group, we took a primary care perspective on the management of CKD. Our search was extensive and comprehensive, with no restrictions on language or publication type. By using the ROM as the effect measure for continuous outcomes, we were able to pool data on different measures in a single analysis and overcome the limitations of the more established SMD. This provided an overall indication of the treatment effect, as did pooling data in cases where there were few studies or a high degree of heterogeneity. The generalisability of our findings is limited by the paucity of studies, the high degree of statistical heterogeneity, and clinical heterogeneity in terms of comorbidities and treatment interventions. We concluded that clinical heterogeneity was inevitable in studies in these patients, as most were undergoing treatment with background medications. Paucity of data restricted our ability to explore the baseline characteristics of studies and the effects of lipid-modifying drugs, glycaemic control drugs and bicarbonates, particularly on proteinuria and our other secondary outcomes. We were also restricted in our ability to investigate the link between treatment effect and drug dose and assess publication bias fully. There was a wide variation of study characteristics (in terms of population size and follow-up period) and baseline characteristics (in terms of male/female split, smoking status, ethnicity and hypertensive status). The generalisability of the findings is also limited by the effect of drugs on CKD being secondary in all the studies. For example, antihypertensives target hypertension and glycaemic control drugs target diabetes. Our ability to assess the quality of the included studies was restricted, because for nearly half the studies we could not assess allocation concealment bias, and for around a quarter of the studies we could not ascertain the risk of attrition bias or whether the participants, personnel and outcomes assessors had been blinded. Further bias may have arisen as most of the reports in our included studies

were subgroup or post-hoc analyses of trials. The balance achieved by randomising the whole trial population may not apply to subgroups of patients. However, we were unable to check this as few of these studies reported the necessary baseline characteristics, split by intervention group. Few studies reported the use of maintenance dialysis and kidney transplantation, which may be considered to be more appropriate survival outcomes for patients with CKD stages 3 and 4, compared with mortality. Limitations may also arise from the possible impact on patient management decisions following the introduction of CKD staging in 2002. Before that, patients were treated on the basis of the perceived cause of CKD rather than the CKD itself. A number of the included studies had recruitment periods that began before 2002, so their populations would be a mixture of patients managed under the old and new guidance. However, in spite of all these limitations, the results of our analyses are consistent with those of other studies.

### Implications for clinical practice and future research

In populations with CKD stages 3 and 4, lipid-modifying drugs and glycaemic control drugs may improve renal function, and lipid-modifying drugs are associated with a reduction in cardiovascular events. Although antihypertensive drugs and sodium bicarbonate are presumed to reduce CKD progression,[8 11–13] we did not find a significant treatment effect. However, the paucity of data and high heterogeneity do not allow strong recommendations for clinical practice. Therefore, future research is needed into the effects of drugs on the progression in CKD, in particular in studies that focus on CKD progression as a primary outcome.

Our finding of limited data is important given that CKD is a common progressive disease. The drugs we considered are commonly used, but studies have concentrated on cardiovascular outcomes and not CKD outcomes.

## CONCLUSIONS

This review suggests that glycaemic control and lipid regulation may improve renal function in patients with moderate CKD. There was no pooled evidence of either benefit or harm from antihypertensive drugs. However, the data were limited and heterogeneity was high for many of the analyses. Therefore, our findings are tentative.

**Author affiliations**
[1]Nuffield Department of Primary Care Health Sciences, University of Oxford, Oxford, UK
[2]Department of Public Health and Primary Care, KU Leuven, Leuven, Belgium
[3]Institute of Applied Health Research, University of Birmingham, Birmingham, UK
[4]Bodleian Health Care Libraries, University of Oxford, Oxford, UK
[5]John Radcliffe Hospital, Oxford Radcliffe Hospitals Trust, Oxford, UK
[6]Farr Institute of Health Informatics Research, University College London, London, UK

**Acknowledgements** We would like to thank all those who clarified our queries about their trials and/or provided additional data, including Professor Nathaniel Hawkins, Professor Fan Fan Hou, Ms Jennifer Meessen, Professor Bruce Neal, Professor Valdo Perkovic, Dr Ruth Peters, Professor Scott D Solomon, Dr Di Xie and a number of anonymous others. We would also like to acknowledge all those who provided data and/or clarification about their trials which, unfortunately, we were unable to include in our study, including Professor Fan Fan Hou and Dr Di Xie, Mr Colin Bicknell, Professor Bob Byington, Dr Enayet Karim Chowdhury, Professor Fan Fan Hou, Dr Dipak Kotecha, Dr Javier Martinez-Martin, Professor John McMurray, Dr Robert G Nelson, Nove Nordisk, Hiromi Rakugi MD, Dr Marcello Tonelli, Professor William White, Dr Di Xie and Professor Keng Thye Woo. Thanks also to Dr Boby Mihaylova, who helped produce the protocol for this review; Dr Richard Stevens, who provided some statistical advice; Dr Brian Shine, who provided advice on pooling different measures of CKD and proteinuria; Professor Paul Glasziou, Dr Boby Mihaylova, and from the Patient and Public Involvement Group, Dr Elizabeth Holloway, Dr Peter Kirby and Mrs Rachael Patel, for their comments about the manuscript and helpful suggestions that have improved the text; and the Stakeholder Group and Steering Committee of our NIHR Programme Grant, for discussions that inspired this paper.

**Contributors** JM, RP, CRB, JYV, JKA, DSL, CAOC, KT, FDRH and AB were involved in the conception and design of the study. NR, assisted by JM and JKA, devised the search strategy. NR completed the search. JM, NP, KST, JYV and SF selected studies for inclusion, and KST, JM, JYV and NP extracted data. KST carried out the data analysis with advice from RP. KST and JM together wrote the first draft of the manuscript, and all authors contributed critically to subsequent revisions and approved the final manuscript. KST and JM had full access to all data in the study and take responsibility for the integrity and accuracy of the data analysis. KST and JM are guarantors.

**Funding** This article presents independent research funded by the NIHR under the Programme Grants for Applied Research (RP-PG-1210-12003). The views expressed are those of the authors and not necessarily those of the NHS, the NIHR or the Department of Health.

**Competing interests** KST, JM, JKA and SF receive funding from the NIHR Programme for Applied Research. JYV receives funding from the NIHR Community Healthcare MIC, Nuffield Department of Primary Care Health Sciences and University of Oxford NIHR Diagnostic Evidence Co-operative (DEC). DSL receives funding from the NIHR Community Healthcare MIC. FDRH acknowledges his part-funding from the National Institute for Health Research (NIHR) School for Primary Care Research, the NIHR Collaboration for Leadership in Health Research and Care (CLAHRC) Oxford, the NIHR Oxford Biomedical Research Centre (BRC) (UHT), and the NIHR Oxford Medtech and In Vitro Diagnostics Co-operative (MIC). RP receives funding from the NIHR Oxford Biomedical Research Centre Programme, the NIHR Programme for Applied Research, the NIHR HPRU Gastrointestinal Infections Group, and the NIHR Diagnostic Evidence Co-operative (DEC).

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
