## [Reviewer comments · BMJ Open]

ARTICLE DETAILS

TITLE (PROVISIONAL)	Effects of antihypertensives, lipid-modifying drugs, glycaemic control drugs, and sodium bicarbonate on the progression of stages 3 and 4 chronic kidney disease in adults: a systematic review and meta-analysis
AUTHORS	Taylor, Kathryn; Mclellan, Julie; Verbakel, Jan; Aronson, Jeffrey; Lasserson, Daniel; Pidduck, Nicola; Roberts, Nia; Fleming, Susannah; O'Callaghan, Christopher; Bankhead, Clare; Banerjee, Amitava; Hobbs, Richard; Perera, Rafael

VERSION 1 – REVIEW

REVIEWER	Wubshet Tesfaye University of Tasmania, Australia
REVIEW RETURNED	18-Apr-2019

GENERAL COMMENTS	The authors have done a commendable job addressing one of the most important issues in CKD patients - the effect of different potentially renoprotective medications and their relationship with disease progression. I have only minor comments: 1. Can the authors be consistent in their use of numbers. For example, in page 13 lines 23-57, some of the numbers were spelled out while others were written in numbers.2. On result tables 2 and 3 the authors attached as supplementary files, can you please describe the years the researches were published besides the author/study name? I see some of them were mentioned, but not in all. Especially, if you put the years on Table 2 it will be easy for readers to look for the papers.
--

REVIEWER	Liming Li Professor, School of Public Health, Peking University, China
REVIEW RETURNED	30-Jun-2019

GENERAL COMMENTS	For the statistical part of this review, I have several comments. 1. It seemed that the reviewers combined studies with different control (placebo, no drug intervention, or a drug from the three other classes) together. This was not reasonable as they were addressing different effects.2. The reviewers used ROM to combine studies reporting different measures of renal function or proteinuria. However, considering the differences among these outcome measures, the combination was not encouraged in usual practice of systematic reviews. This would also contribute to increased heterogeneity.3. Page 11 of 71, Line 44. Please provide the definition of high or low blood pressure.
---

	4. It may be more reasonable to consider maintenance dialysis and kidney transplantation as a survival outcome.
--	---

REVIEWER	Dr. Ferrán Catalá-López National School of Public Health, Institute of Health Carlos III, Madrid, Spain
REVIEW RETURNED	01-Jul-2019

GENERAL COMMENTS	This manuscript reports the methods and results of a meta-analysis synthesizing evidence about the effects of antihypertensives, lipid-modifying drugs, glycaemic control drugs and sodium bicarbonate on the progression of stages 3 and 4 chronic kidney disease in adults. With moderate or (very) high heterogeneity, the authors were able to conclude that (a) glycaemic control and lipid-modifying drugs may slow the progression of CKD, but (b) no evidence of benefit nor harm from antihypertensive drugs. General comments: This manuscript was a pleasure to read. The authors did a thorough systematic review and meta-analysis of the relevant literature. Unfortunately, the literature is sparse and very heterogeneous (most notably, treatment interventions). While they found 35 eligible studies, the outcomes only overlapped in a few of those studies. While it was possible to draw some conclusions about estimated glomerular filtration rate (9 studies for antihypertensive therapy, 5 studies for lipid-lowering therapy, 6 glycaemic control therapy, 2 studies sodium bicarbonate), all other outcomes were only informed by data from 3 or fewer studies (often just 1 to 3 studies). For example for the following outcomes, proteinuria (3 studies only for antihypertensive therapy), adverse events (1 study for antihypertensive therapy, 3 studies for lipid lowering therapy, 1 study for glycaemic control therapy), dialysis/kidney transplantation (1 study for antihypertensive therapy, 3 studies for lipid-lowering therapy, 1 study for glycaemic control therapy), cardiovascular events (2 studies for antihypertensive therapy, 2 studies for lipid-lowering therapy, 2 studies for glycaemic control therapy), or all cause mortality (3 studies for antihypertensive therapy, 2 studies for glycaemic control therapy, ...). This made it difficult or impossible to draw strong conclusions about the effects of these therapies on the progression of stages 3 and 4 chronic kidney disease in adults. Specific comments/questions: Page 7. Line 39. Description of treatment interventions. The authors' state that interventions of interest and comparators were: "(1) drugs that may modify the progression of CKD, antihypertensives, lipid-modifying drugs, glycaemic control drugs, and sodium bicarbonate; (2) Control arms were either given a placebo drug, no drug intervention, or a comparator drug from the three other classes". Could you please further describe the interventions and comparators? For example, there are many classes of antihypertensives, which lower blood pressure by different means. Among the most important and most widely used drugs in chronic kidney disease are ACE inhibitors, ARBs, thiazide diuretics, but also other medications (such as calcium channel blockers, and beta blockers). In addition, I think authors should clearly describe inclusion and exclusion criteria for treatment regimens and should provide justification when treatments are merged to form single
---

	comparators (a practice sometimes described as “lumping” of interventions). Often, one has to decide whether to lump or split treatments—that is, whether to combine multiple drugs of the same therapeutic class, or different doses of the same drug, or varying durations of administration, or different controls. In my opinion, lumping requires treatments to have similar treatment effects, and although this technique is appropriate in some cases, it should be supported by a clear rationale when performed. Authors should describe the included treatments and adherence to (and assessment of) the homogeneity assumption. For example (p.57, Supplementary Table 3), glycaemic control drugs are represented by 1 single trial of gliptins and 5 trials of flozins, but no trials of other antidiabetic drugs. Lipid-lowering drugs considered trials on statins, fibrates, or their combination (alone, with diet...). Antihypertensives included ACEi/ARBs, but also mineralcorticoid receptor antagonists such as spironolactone and eplerenone. Could you please clarify why? I think understanding this is key considering their conclusion “Glycaemic control and lipid-modifying drugs may slow the progression of CKD, but we found no evidence of benefit nor harm from antihypertensive drugs.” Minor comments: Page 6. Methods. Lines 36 and 37. The authors' state: “Our systematic review was registered and conducted in line with recommendations for systematic reviews and meta-analyses (PRISMA).” In my opinion, the authors should rephrase this in order to clarify they used PRISMA reporting guidance (not conducting). To clarify, PRISMA is not intended to be prescriptive about how systematic reviews and meta-analyses should be conducted/performed or interpreted. Instead, PRISMA seeks to provide reporting guidance on important information to be included in reports of systematic reviews and meta-analyses. Page 6. Methods. Line 36. I would mention a protocol exists. For example: “We developed a systematic review protocol and registered with PROSPERO (registration number: CRD42015017501)”. Please, describe how the strength of the body of evidence was assessed (such as GRADE)
--	---

VERSION 1 – AUTHOR RESPONSE

Reviewer 1:	
Can the authors be consistent in their use of numbers? For example, in page 13 lines 23-57, some of the numbers were spelled out while others were written in numbers.	We have amended the text to adhere to BMJ house style, except in two situations:  1. If a number starts a sentence where we have all numbers in full 2. Where we are referring to chronic kidney stages 1 – 5 as numbers are the widely accepted format for these terms BMJ house style states “Numbers under 10 are spelt out, except for measurements with a unit (8 mmol/l) or age (6 weeks old), or when in a list with other numbers (14 dogs, 12 cats, 9 gerbils)”
On result tables 2 and 3 the authors attached as supplementary files, can you please describe the years the researches were published besides the author/study name? I see some of them were mentioned, but not in all. Especially, if you put the years on Table 2 it will be easy for readers to look for the papers.	We have added a new column to both of these tables, which are now in the supplementary material file (SuppTable3 and SuppTable4), to show the year(s) of publications for each study.
Reviewer 2:	
It seemed that the reviewers combined studies with different control (placebo, no drug intervention, or a drug from the three other classes) together. This was not reasonable as they were addressing different effects.	We thank the reviewer for highlighting this point. Most of the control groups in our included studies were given placebos. We have added new text in our results section to clarify the numbers of studies that gave their control groups non-placebo drugs. Excluding these studies did not alter the conclusions, as shown in the results in our supplementary material (Supplementary material: Table 6).
The reviewers used ROM to combine studies reporting different measures of renal function or proteinuria. However, considering the differences among these outcome measures, the combination was not encouraged in usual practice of systematic reviews. This would also contribute to increased heterogeneity.	Our study is a first attempt to summarise the literature on drug treatment for patients with CKD types 3 and 4. We accept that there are limitations in terms of high statistical and clinical heterogeneity and paucity of studies. We used the ROM as it enabled us to pool more data and give an “overall indication of the treatment effect”, but we realise that our findings can only be tentative given the data limitations. Pooling data from different measures, using the standardised mean difference (SMD), is an established approach, which is recommended by the Cochrane Handbook, and the GRADE guidelines highlight how using the ROM overcomes some of the limitations of SMD. Estimated ROMs of final eGFRs were calculated for changes and rates rather than endpoints (different measures) and, as the results in our supplementary material file shows, removing these estimates produced some reductions in statistical heterogeneity (suggesting that including these studies increased heterogeneity), but this does not invalidate our approach. We investigated changes in

heterogeneity and found that removing the study with estimates of final eGFR from the lipid regulating study (SHARPb) reduced the I-squared from 88.3% to 45.3%, reflecting a marked reduction in statistical heterogeneity. Although statins act via the same mechanism, adding these SHARP data may have increased the clinical heterogeneity, as the intervention was a statin (as with other studies), but at a higher dose and combined with another lipid modifying drug. We concluded that clinical heterogeneity was inevitable in trials of these patients as the majority were taking background medication. There were too few proteinuria studies to comment on the changes in statistical heterogeneity in the sensitivity analysis.

In response to this reviewer's comments we have provided further clarification and have increased the transparency of our reporting by adding new text. First, we have added the following to the methods section:

"Using the ROM effect measure overcomes limitations of the well-established standardised mean difference, by providing units that are more easily interpreted than standard deviations (SDs), and avoiding the problems associated with presenting results with SDs, which can produce misleading results by deflating treatment effects for studies of heterogeneous populations and inflating treatment effects in homogeneous populations."

We have added the following to the results for studies of lipid modifying drugs:

"The high degree of heterogeneity was attributed to the SHARP study¹⁰¹⁻¹⁰³ of the CKD stage 4 subgroup, as removing this study reduced the I² from 88.3% to 45.3%. The intervention for this study was a higher dose of statin than in the other studies, and another lipid modifying drug, ezetimibe, was also used."

In the strengths and limitations section we have added, for strengths:

"Our study is a first attempt to summarise the literature on drug treatment for patients with CKD types 3 and 4 ... By using the ROM as the effect measure for continuous outcomes we were able to pool data on different measures in a single analysis and overcome the limitations of the more established SMD. This provided an overall indication of the treatment effect, as did pooling data in cases where there were few studies or a high degree of heterogeneity."

	And for limitations, we have added “.....and treatment interventions. We concluded that clinical heterogeneity was inevitable in studies of these patients, as most were undergoing treatment with background medications.” We have also now included GRADE assessments to formally report the low level of confidence in our estimates, and our final sentence of the conclusions section now reads “Therefore, our findings are tentative.”
Page 11 of 71, Line 44. Please provide the definition of high or low blood pressure.	We have added definitions. This refers to a pre-specified subgroup analysis that was not carried out, as there were insufficient studies.
It may be more reasonable to consider maintenance dialysis and kidney transplantation as a survival outcome.	We thank the reviewer for highlighting this point. Our limitations sections now includes the following text: “Few studies reported the use of maintenance dialysis and kidney transplantation, which may be considered to be more appropriate survival outcomes for patients with CKD stages 3 and 4, compared with mortality.”
Reviewer 3:	
This manuscript was a pleasure to read. The authors did a thorough systematic review and meta-analysis of the relevant literature.	We thank the reviewer for these positive comments.
Unfortunately, the literature is sparse and very heterogeneous (most notably, treatment interventions). While they found 35 eligible studies, the outcomes only overlapped in a few of those studies. While it was possible to draw some conclusions about estimated glomerular filtration rate (9 studies for antihypertensive therapy, 5 studies for lipid-lowering therapy, 6 glycaemic control therapy, 2 studies sodium bicarbonate), all other outcomes were only informed by data from 3 or fewer studies (often just 1 to 3 studies). For example for the following outcomes, proteinuria (3 studies only for antihypertensive therapy), adverse events (1 study for antihypertensive therapy, 3 studies for lipid lowering therapy, 1 study for glycaemic control therapy), dialysis/kidney transplantation (1 study for antihypertensive therapy, 3 studies for lipid-lowering therapy, 1 study for glycaemic control therapy), cardiovascular events (2 studies for antihypertensive therapy, 2 studies for lipid-lowering therapy, 2 studies for glycaemic control therapy), or all cause mortality (3 studies for antihypertensive therapy, 2 studies for glycaemic control therapy, ...). This made it difficult or impossible to draw strong conclusions about the effects of these therapies on the progression of stages 3 and 4 chronic kidney disease in adults.	We now include GRADE assessments to formally report the low level of confidence in our estimates. We have added further text to clarify the strengths and limitations of our study, acknowledging that we have only given an “overall indication of the treatment effect” and tentative findings. On the subject of clinical heterogeneity, we have now stated: “The generalisability of our findings is limited by the paucity of studies, the high degree of statistical heterogeneity, and clinical heterogeneity in terms of co-morbidities and treatment interventions. We concluded that clinical heterogeneity was inevitable in studies in these patients, as most were undergoing treatment with background medications.”

Page 7. Line 39. Description of treatment interventions. The authors' state that interventions of interest and comparators were: "(1) drugs that may modify the progression of CKD, antihypertensives, lipid-modifying drugs, glycaemic control drugs, and sodium bicarbonate; (2) Control arms were either given a placebo drug, no drug intervention, or a comparator drug from the three other classes". Could you please further describe the interventions and comparators? For example, there are many classes of antihypertensives, which lower blood pressure by different means. Among the most important and most widely used drugs in chronic kidney disease are ACE inhibitors, ARBs, thiazide diuretics, but also other medications (such as calcium channel blockers, and beta blockers).	We have added the drugs list to our supplementary material file (Supplementary material: Table 2) showing the four drug classes, drug type (e.g. ACE, ARB, and statin), and named drugs. In addition, there is now an extra column in the table of interventions in our supplementary material file (Supplementary material: Table 4) to show the drug type. We pre-specified subgroup analysis by drug type within each drug class, but we were only able to report the data for the primary outcome for antihypertensive drugs (supplementary material: Figure 4).
In addition, I think authors should clearly describe inclusion and exclusion criteria for treatment regimens and should provide justification when treatments are merged to form single comparators (a practice sometimes described as "lumping" of interventions). Often, one has to decide whether to lump or split treatments—that is, whether to combine multiple drugs of the same therapeutic class, or different doses of the same drug, or varying durations of administration, or different controls. In my opinion, lumping requires treatments to have similar treatment effects, and although this technique is appropriate in some cases, it should be supported by a clear rationale when performed. Authors should describe the included treatments and adherence to (and assessment of) the homogeneity assumption. For example (p.57, Supplementary Table 3), glycaemic control drugs are represented by 1 single trial of gliptins and 5 trials of flozins, but no trials of other antidiabetic drugs. Lipid-lowering drugs considered trials on statins, fibrates, or their combination (alone, with diet...). Antihypertensives included ACEi/ARBs, but also mineralcorticoid receptor antagonists such as spironolactone and eplerenone. Could you please clarify why? I think understanding this is key considering their conclusion "Glycaemic control and lipid-modifying drugs may slow the progression of CKD, but we found no evidence of benefit nor harm from antihypertensive drugs."	The four drug classes arose from our literature and scoping reviews, as they are known to modify the rates of progression of CKD. These two reviews are outlined in our protocol. We now clarify that our protocol was registered with PROSPERO (see below). We recognised that there were many drugs within each drug class used in clinical practice by pre-specifying subgroup analysis by drug type. Our pre-specified subgroups also included the intensity of the intervention, by dose and dosage regimen, but, as we have reported, there were insufficient studies to carry out these subgroup analyses.
Page 6. Methods. Lines 36 and 37. The authors' state: "Our systematic review was registered and conducted in line with recommendations for systematic reviews and meta-analyses (PRISMA)." In my opinion, the authors should rephrase this in order to clarify they used PRISMA reporting guidance (not conducting). To clarify, PRISMA is not intended to be prescriptive about how systematic reviews and meta-analyses should be conducted/performed or interpreted. Instead, PRISMA seeks to provide reporting	We thank the reviewer for highlighting this point. We have reworded the manuscript text to read: "Our systematic review protocol was registered with PROSPERO and reported in line with recommendations from the PRISMA statement for reporting systematic reviews and meta-analyses of studies that evaluate healthcare interventions."

guidance on important information to be included in reports of systematic reviews and meta-analyses.	
Please, describe how the strength of the body of evidence was assessed (such as GRADE).	We have completed a GRADE assessment of the certainty of the evidence for the primary outcome and we have now included the summary of findings table in our supplementary material (Table 7).

VERSION 2 – REVIEW

REVIEWER	Liming Li Professor, Department of Epidemiology and Biostatistics, School of Public Health, Peking University Health Science Center, China.
REVIEW RETURNED	14-Aug-2019

GENERAL COMMENTS	The authors have made substantial revisions according to the comments, or have acknowledged relevant limitations in the conduct of this systematic review.
--

REVIEWER	Dr Ferrán Catalá-López National School of Public Health, Madrid, Spain
REVIEW RETURNED	14-Aug-2019

GENERAL COMMENTS	Thank you for inviting me to review this revised version of the manuscript. Overall, the authors have addressed all my previous questions/concerns. I have no further suggestions.
---